# Healthy Lifestyle Behavior, Goal Setting, and Personality among Older Adults: A Synthesis of Literature Reviews and Interviews

**DOI:** 10.3390/geriatrics7060131

**Published:** 2022-11-23

**Authors:** Ming Yu Claudia Wong, Kai-ling Ou, Pak Kwong Chung

**Affiliations:** 1Department of Health and Education, The Education University of Hong Kong, Hong Kong, China; 2Department of Sport, Physical Education and Health, Hong Kong Baptist University, Hong Kong, China

**Keywords:** healthy lifestyle behavior, goal setting, personality, older adults, qualitative interview

## Abstract

Background: Despite the well-known health benefits of adopting a healthy lifestyle, older adults’ self-determination, goals, and motivation, as well as other personality factors, are known to influence their healthy lifestyle behaviors, yet these interactions have rarely been discussed. Method: The literature that investigated and discussed the interaction of personality, goals, and healthy lifestyle behaviors among older adults was reviewed. In addition, interview responses from older adults regarding their experiences in participating in a real-life physical activity intervention and its relationship with their personality traits and goal setting were synthesized using content analysis. Results: The current review highlights the relationship between healthy living practices, goal setting, and personalities, and it is backed up and expanded upon by interviews with participants. People with different personality types are likely to have diverse views on HLBs. Individuals who are more conscientiousness or extraverted are more likely to adopt HLBs than those who are not. Discussion: It is suggested that a meta-analysis should be conducted on the relationship between personality, goal setting, and physical exercise or other specific HLBs. In addition, future research should focus on various types of HLB therapies that take into account personality and goal setting.

## 1. Highlights

The current review highlights the relationship between healthy living practices, goal setting, and personality.Conscientiousness, neuroticism, and extraversion were shown to have significant effects on health-related behaviors as well as actual healthy lifestyle behaviors.Future research should focus on various types of HLB therapies that take into account personality and goal-setting processes.

## 2. Background

Despite the well-known benefits of engaging in a healthy lifestyle [1,2,3], older adults are expected to have a less goal-oriented mindset than teenagers and younger adults. This is because older adults have a weaker sense of pursuing life goals or they have a scheduled routine for their life that facilitates self-regulation. Therefore, older adults’ self-determination, goals, and motivation to engage in a healthy lifestyle, as well as other personality factors, are seen to influence their healthy lifestyle behaviors. A few well-documented factors have been proven to have an impact on older adults’ adherence to a healthy lifestyle or engagement in physical activities in particular, including believing in the benefits of exercising and having experience with exercise, setting goals, and having certain personality traits [4]. Considering the unchangeability of older adults’ past experiences, setting appropriate goals based on individual personalities is expected to be useful in fostering older adults’ participation in physical activity or adherence to other healthy lifestyle behaviors. The existing literature describes the relationship between personality and goals among older adults, with their outcomes associated with subjective well-being, quality of life, and other mental benefits [5,6]. Previous research also indicated that various sociocultural variables, such as gender, education level, the sense of autonomy, and the physical capability of older adults, should be taken into account when determining their attitudes toward and intention to participate in physical activities. It is believed that various factors will negatively affect older adults’ personality, autonomy, and awareness of physical activity goal setting, thus affecting their healthy lifestyle behaviors [7,8].

According to the literature, life goals are considered as an internal mental conception of desired outcomes or activities that a person aims to pursue in their daily life [9]. Based on the self-determination theory, individuals emphasize the importance of goal content rather than only having objectives and claim that these types of goals are important. Personality is seen as a factor influencing the construction of goals [10]. Psychologists view personality traits and life goals as part of a distinct concept or theory, with personality traits being comparatively stable and consistent, whereas life goals might fluctuate due to life changes. Other psychologists [11,12] claim that goals or life tasks are the “doing” or “acting” side of the personality. Moreover, goals are described as dynamic components of personality that reflect one’s interaction with the environment across time. As a result, goals serve as a link between personality traits and behavior. Based on establishing the concept of goals as personality-in-context [13], a research study showed that extraversion was associated with understanding the importance of intrinsic goals, having good health, and making progress in social goals among older adults [6,14]. Conscientiousness results in higher levels of health and social progress, while neuroticism only results in lower progress toward social goals and no progress in health goals [14]. Furthermore, individuals with higher neuroticism are more sensitive to stress and have significantly moderate progress in health and social goals, while conscientiousness individuals who are sensitive to stress have significantly moderate progress only in health goals, and for extraverts, there is no moderation effect on any goal progress [14]. Nevertheless, optimism is associated with goal attainment and continuity and leads to good health [5,15]. Unsurprisingly, goal continuity showed a negative correlation with neuroticism, and a positive correlation with conscientiousness [15]. However, a research study also claimed that the relationship between goal orientation and proactive health-related coping behavior depends on the stressors, yet this has not been completely explained by individual differences [16].

Although the relationship between goals and personality has been investigated in previous studies, to the best of our best knowledge, the relationship between personality and particular health goals among older adults has been less discussed. With regard to the importance of healthy and active aging, pursuing health-related goals is essential for older adults. Meanwhile, the interaction between personality and appropriate health goals among older adults is not yet known. Therefore, a discussion on the interaction of personality, goals, and healthy lifestyle behaviors among older adults should be presented. 

## 3. Objectives

The purpose of this review is to emphasize the importance of personality and goal setting in older adults’ participation in healthy lifestyle behaviors. The body of research on this topic has been limited; therefore, as part of this study, we reviewed the prior literature to examine the interactions between personality, goals, and healthy lifestyle behaviors among older adults. We synthesized the literature review to address the first research question (RQ1): How does the interaction between personality, goals, and healthy lifestyle behaviors (e.g., physical activity and eating habits) manifest among adults? Second, we analyzed interviews with older adults who participated in the authors’ previous physical activity intervention program to determine how their participation was associated with their personality traits and goals. To achieve this objective, we formulated the second research question (RQ2): How do interviewees describe their most satisfying goals for facilitating participation in healthy lifestyle behaviors? 

## 4. Method

### 4.1. Literature Search

The two authors, a post-doctoral research fellow and a PhD student, each with extensive knowledge on physical health and developmental psychology, conducted a literature search on the body of research that investigates the relationship between personality and health-related goals. The Scopus and Web of Science databases were searched using the keywords (“Older Adults” AND “Goals” AND “Personality) AND (“Healthy Lifestyle Behaviour” OR “Physical Activity” OR “Eating Habits”). The extracted research articles were imported to EndNote for management. After removing duplicates, the authors screened the titles and abstracts of the retrieved studies independently to identify relevant research articles. Disagreements were settled through discussions. Considering the generalizability of the literature synthesis covering adults to young, old, and older adults, the only inclusion/exclusion criterion for screening relevant papers was to exclude studies that targeted the samples of subjects aged 18 or below. Studies that mentioned older adults were also included. Appropriate papers were then identified and processed for a full-text review. The literature synthesis mainly targeted the following information: (1) types of personality traits, (2) types of goals, (3) content of goals, (4) types of healthy lifestyle behaviors, and (5) how they interact. 

### 4.2. Interview 

A total of 20 participants from the authors’ previous intervention program [17] were invited to participate in semi-structured in-depth interviews after completing the program. These 20 people participated in either tai chi or resistance training interventions during three 1 h sessions per week over 18 weeks. Questions addressed participants’ goals in the intervention program and for daily healthy living, their perceived impact of personality on goal setting, forms of preferred goals, and the attractiveness of extrinsic rewards. The interview guide included the following questions: 1. Did you set any goals for yourself before this program? What were they?2. Think about your overall health and fitness and ability to get around and do the things you want to do. What are your wishes and hopes for that in the future?2a. Do you think this program/participating in physical activity has facilitated you to do so, or at least achieve a certain extent of that?3. Again, think about your health and fitness and ability to get around and do the things you want to do. What are your fears and worries about the future?3a. Do you think this program/participating in physical activity could eliminate these fears or worries?4. What kinds of goals do you tend to set for yourself, in terms of physical activity/health-related factors (e.g., diet, exercise)?5. What kind of personality do you think you have?5a. Do you think your personality affects your goals for physical activity and overall health? How?6. Do you prefer having standardized static goals or personalized goals when participating in an exercise program with coach supervision?7. Do you prefer having a single ultimate goal or integrated smaller goals (goal phrase) before achieving the ultimate goal?8. To what extent could rewards prompt you to achieve your goals?

### 4.3. Data Analysis

The literature review was synthesized and interview responses were analyzed by first creating an open-coding label to identify relevant information and factors, individual differences, types of goals, and types of healthy lifestyle behaviors, as well as potential interactions between factors. The above-mentioned study objectives provided the basis for the initial categorization. Then, the relevant open codes were refined and combined by labelling with an “analytical” theme. After this process, the second coder went through the same process together, expressing agreement with the first coder’s labels, indicating inter-rater reliability. The audio recordings of the interviews were transcribed verbatim and translated from Chinese to English. The verbatim transcripts were screened by another author to eliminate translation variations, strengthening the qualitative research’s credibility. The data were managed by using NVivo 12 (QSR, 2020) using memo writing, coding, and categorizing until data saturation was reached. A similar inter-rater reliability process was used for analyzing the interview’s transcripts. 

## 5. Results 

### 5.1. Literature Review and Interview Demographics

A total of 22 research articles were identified during the database search. Figure 1 displays the flowchart of the search. The research articles that were retrieved and included in the synthesis mentioned the effects of personalities on healthy lifestyle behavior (HLB)-related goals among older adults as well as younger adults. A total of 20 people participated in the in-depth interviews:17 women and 3 men; the age range for 17 of them was 65–69 years, and for 3 of them, it was 70–74 years. The inter-rater reliability showed 95% agreement in both the literature synthesis and interview transcript analysis. 

#### 5.1.1. Types of Healthy Lifestyle Behavior 

Most of the 22 research articles mentioned the types of healthy lifestyle behavior that may be affected by personality traits and goal achievement. They include exercising, participating in sports, eating a healthy diet, participating in physical activity using electronic devices [19,20], participating in leisure-time physical activity that is “planned, structured, repeated and with maintenance” [21], controlling one’s weight, managing one’s nutrition [22], engaging in social activities, and having intentions and beliefs about physical activity [23]. The literature indicated that people who engage in these HLBs tend to have better physical, mental, emotional, and cognitive health [24]. 

#### 5.1.2. Personality Types and Healthy Lifestyle Behaviors 

Among the 22 research articles, personality traits were mostly represented by the “Big Five” traits [21,23,25,26,27,28,29,30,31] and measured by the Big Five Inventory [32]. Research has shown that personality traits play an important role in influencing people’s health and HLB choices throughout their lives [28]. Studies have shown that people who are highly extraverted, highly conscientious, and not neurotic tend to have higher levels of self-efficacy, self-motivation, and self-control [21] and, thus, engage in more leisure-based physical activities. An intervention study showed that for people with a high level of conscientiousness, their step count was positively predicted from a pre-test, while under monitoring during the intervention, people with a high level of neuroticism showed a significant increase in their daily step counts as well [30]. Meanwhile, people with a sense of openness tended to be slightly more active, but there were no significant differences in the level of leisure-based physical activity among people with agreeableness [31,33,34]. In addition to people with high neuroticism, people with other types of personality traits were associated with a high level of engagement in leisure activities, including social and physical activities, which indicates successful active aging [29]. 

Other than the Big Five personality traits, the relationship between conscientiousness and HLB was shown using the effect of perfectionism [35,36] and measured using self-oriented and socially prescribed perfectionism subscales [37]. Despite its association with a high level of self-discipline, perfectionism was also associated with extreme behaviors such as binge eating, which is considered to have a deleterious effect on physical and mental health [36]. The opposite of perfectionism, according to Sirois [24], may be chronic procrastination, which is defined as a trait-like personal characteristic. It was shown to have an association with difficulties in self-regulation and the avoidance of engaging in HLBs such as physical activity and healthy eating habits, thus resulting in poor physical health and well-being [38,39,40]. Some research studies [41,42] discussed personalities in a more general manner by dividing them into optimistic and pessimistic, as measured by the Life Orientation Test Revised (LOT-R) [43]. They showed that optimistic individuals tend to have a higher mental health-related quality of life [42], as well as favorable physical health outcomes [44,45]. Another study [22] investigated personality measurements in terms of the Type D personality, which is correlated with obesity, unhealthy lifestyle behavior, and vulnerability [22,46].

#### 5.1.3. Healthy Lifestyle Behavior, Goals, and Personality Traits 

There are multiple types of goals, including achievement, maintenance, disengagement, engagement, and compensation goals [47], and the retrieved research studies mostly focused on achievement and disengagement goals. Achievement goals were examined and applied mostly in the context of healthy lifestyle behavior interventions, which consisted of physical activity, nutrition management, stress management, and cognitive function training. Achievement goals in interventions tended to be mutual goals regardless of personality traits [22]. On the other hand, research on M-health indicated that electronic devices, such as a Fitbit, were considered to be effective in facilitating goal setting and achievements. This is because the planning, learning [19], rewarding, comparing, and sharing functions in electronic devices can trigger individuals’ motivation to achieve their desired goals [30,48]; it is also noteworthy that electronic devices can support personalized goals based on individual personality traits. 

Engagement goals involve more inflexible pursuits, and research has shown that disengagement goals with more flexibility (e.g., short-term or long-term goals) for adjustment were also important for achieving a healthy life [49,50]. Apart from theoretically supported goals, qualitative research has also revealed other forms of desire goals or preventive goals related to achieving HLBs, which were labelled as “hoped-for possible self” and “feared possible self” [24]. The “hoped-for” goals include being physically active, being a vegetarian, reducing weight, etc. Fear-related goals include preventing oneself from becoming obese, preventing heart disease, preventing weak lower limbs, etc. These kinds of goals can also encourage individuals to engage in HLBs. They are also considered to be similar to desired goals and “anti-goals” [25]. Research has shown that individuals stick with HLBs when they have goal-relevant information, such as a concept of their ideal self (both physically and psychologically) and health-related information [31]. Goals, with meaningful purpose and rewards for achievement, were also considered as effective in achieving a healthy lifestyle. 

Persistent and perceived goal progress can also be influenced by individual personality traits [47]; for instance, people with a higher level of conscientiousness tend to show better self-control and self-regulation and engage in self-corrective actions when pursuing goals [21,51]. Moreover, conscientiousness is strongly associated with a high level of perfectionism, which also correlates with higher personal standards and high-order goals [36], as well as the high-level self-evaluation of goal achievement [26], which might mean vulnerability in terms of the mental status or even health (e.g., eating disorders or extreme eating as related to exercise goals) [35,52]. 

#### 5.1.4. Interaction between Personality, Goals, and Healthy Lifestyle Behaviors

In general, the retrieved studies demonstrated the inter-relationships between personality, goals, and HLBs. Some of the studies also mentioned the lack of a discussion on the role of personality in health-related goal settings and HLBs, which has been shown to have implications in terms of affecting the motivation and perception of HLB among older adults [27,47,50]. 

The studies indicated that personality plays a role in health-related goal setting and HLBs. Only people with a high level of conscientiousness are significantly affected by the product judgments and health-relevant information that they receive, which affects their decision making with respect to dietary choices and physical activity related to their overall goal of a healthy lifestyle [31]. As mentioned above, conscientiousness has been associated with higher levels of self-control, and as discussed in [21], people with self-control tend to give priority to long-term goals over short-term goals, can resist goal-disrupting temptations [53,54], and have strong goal-striving abilities [28], such as avoiding junk food or staying engaged in healthy acts on a daily basis, in order to pursue their long-terms goals consistently. Hence, people with a higher level of self-control, similarly to conscientiousness, are significantly associated with positive subjective well-being and physical activity, mediated by high levels of perceived goal progress and self-efficacy [21,23]. Similarly, Briki and Dagot [25] demonstrated that dispositional self-control and perceived goal progress were negatively associated with neurotic self-attentiveness, thus negatively predicting subjective well-being. Furthermore, people with lower levels of conscientiousness and optimism are associated with low goal re-engagement, higher mental fatigability, and poorer physical and cognitive health maintenance [33]. Meanwhile, optimism and conscientiousness are positively associated with goal achievement, adjustments, and re-engagement and thus are related to older adults’ participating in physical activity and having a healthy quality of life [30]. However, people who exhibit extreme conscientiousness, or perfectionism, tend to overvaluate and think dichotomously about their goal progress, such as weight and body shape, which might lead to unhealthy eating habits or even an eating disorder [35,36]. On the other hand, people who procrastinate tend to have lower expectations for “hoped-for goals” and place no importance on avoiding “feared-for” possible outcomes; thus, they have less intention to change their health behaviors [24]. 

Intervention studies [22] showed that intervention programs that followed an adaptation of goal attainment theories facilitating the development of participants’ mutual goals were able to improve the HLBs of those with Type D personality, reducing their mental vulnerability. Moreover, the vulnerability and established healthy habits of Type D participants were also shown to be influenced by the social support gained from the intervention. Another study [41] demonstrated that people who expressed optimism had higher levels of health-related self-efficacy after the intervention and were able to reduce their waist circumference (lose weight) even one year after the intervention. This indicates that optimistic people have better self-control than pessimistic people with regard to health-related goals and behaviors. 

There was a trend of increasing research investigating the self-determination effect of using electronic health devices, such as smart watches, on goal monitoring and physical activity levels. Studies indicated that even though the functions of these electronic devices could trigger the self-monitoring and self-regulation of HLBs by individuals, including older adults, the research results showed that personality also plays a role in people’s levels of regulation and HLBs. Hence, it is recommended that smart devices provide personalized goal setting systems in which big data can be used to provide goals or HLB plans that are tailored to users’ individual differences and personalities [20]. Bischoff [19] highlighted the importance of goal-specific apps in smart devices in being able to identify individual differences, including personality traits, levels of vulnerability, and usual health behaviors, and being able to provide suitable goal planning and management strategies, as well as relevant features for pursuing HLBs. In addition, based on the differences in physical activity outcomes among people with different personality traits in intervention studies, including personalized interventions and individualized goals based on people’s personality traits, priorities, and attitudes is also suggested, instead of standardized goals and procedures, in order to achieve positive and successful aging [48,55,56]. In addition to individualized goals, promoting programs that offer individual care, verbal encouragement from coaches or instructors, and personalized regular schedules in both research and community settings is suggested to enhance older adults’ HLB [56].

### 5.2. Participant Interviews

#### 5.2.1. Personality

After the interview, all participants were asked to fill in a 16-item personality test based on a framework that evolved from the Myers–Briggs Type Indicator (MBTI) [57]. It indicates personality traits in four dimensions: (1) preferred orientation to life: extraversion (E) or introversion (I); (2) preferred way of perceiving things: sensing (S) or intuition (N); (3) preferred way of making decisions: thinking (T) or feeling (F); and (4) preferred way of dealing with the world: judging (J) or perceiving (P) [58]. 

To be more consistent with the literature synthesis, the relationship between MBTI and the Big Five personality traits was determined. A previous study found that the T-F dimension was correlated with agreeableness and the J-P dimension correlated with conscientiousness; the E-I dimension was strongly correlated with extraversion, and neuroticism was not correlated with any MBTI dimensions [59]. However, when separating the subfactors, researchers [60] later found that E was correlated with extraversion. Openness was significantly associated with N and inversely correlated with S, which involves a direct involvement with information. Agreeableness was negatively correlated with T but positively correlated with F, which means more concern with feelings. Conscientiousness was positively correlated with J and negatively correlated with P, which indicates people who are orderly, deliberate, and self-disciplined. Neuroticism was positively correlated with I but negatively correlated with E; the mental processes of such people are more oriented toward their inner world, which is highly associated with self-consciousness, depression, and anxiety [59].

The results showed that the participants were mainly divided into four types: ESFJ (12), ISFJ (6), INFP (1), and ESTJ (1). According to the findings and combining them with the Big Five personality types, we can clarify that the personality types of these participants are as follows: ESFJ—extroversion, based on information perception, agreeableness, and conscientiousness; ISFJ—introversion (or neuroticism), which relies on information perception, agreeableness, and conscientiousness; INFP—introversion (or neuroticism), openness, agreeableness, and conscientiousness; and ESTJ—extroversion, based on information perception, logical thinking, and conscientiousness. 

#### 5.2.2. Goals and Healthy Lifestyle Behavior 


**Hoped-for possible self**


Participants recalled that before they joined the intervention program, their expectations (goals) for the program were mainly to maintain their physical activity level, improve their physical health, expand their social networks, learn knowledge, and feel a sense of persistence. Only three participants said that they did not set any goals. 

“My goal in retirement is to do more exercises, stay healthy and have fewer doctor visits" (F, R, 10, ISFJ). “Reducing fatty liver” (F, T, 12, ESFJ). “Improve balance and body cold” (F, T,6, ISFJ). “Flexibility in arms and legs and good health” (F, T,14,ESFJ). 

They want to use what they have learned to educate others: “My aim is not only to help myself but also others, as well as teach my friends Tai Chi together” (F, T, 14, ESFJ). Another participant agreed: “I can also teach my family, I will be happy and initiate other older adults” (F, T, 12,ESFJ). 

Some participants wanted to learn about exercise theory to help them maintain their physical activity levels for their entire lives: “I hope that the intervention can open up the path to understanding sports and learn some sports that I can do at home so that I can do them regularly without the help of others” (F, R, 7, ESTJ); “I hope the knowledge of Tai Chi taught by my teacher can be my lifelong asset” (F, T, 2, ESFJ); “I usually do exercise, but I don’t have enough theoretical knowledge, so I sometimes overexert myself and hurt my knee” (F, R, 15, ESFJ). 

One participant noted that perseverance was the goal of program participation: “Try not to miss a single day of training” (F, R, 2, ESFJ).

Those with no goals were mainly inactive and had no sports experience: “I haven’t tried resistance training, I don’t have any goal in doing sports” (F, R, 18, ESFJ); “I don’t have special goals, I usually do stretching at home, I haven’t participated in any other sports activity” (F, R, 19, ESFJ). 


**Fear of possible self**


The worries expressed by participants were mainly having limited physical fitness, becoming injured, and a lack of determination: “I was worried that my muscles were not strong enough or that I was not able to do the movements in the class” (F, R, 11, ESFJ); “I was afraid of Tai Chi standing might hurt my knee” (F, T, 14, ESFJ); “I was afraid I wouldn’t be able to do it [resistance] as I was almost 70 years old and I was afraid it would be too drastic” (F, R, 19, ESFJ); “I was worried that I would give up halfway because I don’t know if the exercise is too difficult” (F, R, 7, ESTJ). 

However, all participants said that the intervention eventually eliminated their concerns. Because some participants believed in the professionalism of the instructors, their stereotypes regarding physical activity changed after learning from the intervention: “Because the instructor is very experienced and knows what age we are, they won’t force us to do it if we can’t” (F, R, 18, ESFJ). A female participant (F, T, 14, ESFJ), after practicing tai chi, said the following: “Standing on one leg in Tai Chi helps me to maintain my balance and I know how to adjust when I fall”.


**Long term health-related goals**


Participants said that they set long-term health-related goals after the intervention and they reported being more conscious of healthy eating habits after participating: “I will pay attention to health information, such as the ingredients of food, to avoid food allergies” (F, R, 10, ISFJ); “I will maintain my weight and take foods that maintain muscle, such as protein” (F, R, 11, ESFJ); “Eat less fried and sweet foods” (M, R, 1, ESFJ); “Eat lighter, less diet out, make breakfast cooked at home” (M, T, 4, ISFJ); “Eat more vegetables, grains and cereals” (F, R, 15, ESFJ).

Regarding physical health, participants stated that the recovering function was their long-term health goal: “I hope my lower limb function can be improved, stronger feet and bones” (F, T, 9, ESFJ); “reducing muscle loss” (F, R, 10, ISFJ). 

Meanwhile, participants had a clearer vision of further goal setting for physical activity. A male participant was determined to use technology to monitor his health: “I will be more aware of different activities. … I use a pedometer to set goals for myself and recently I have been using another function to lift myself up to drink” (M, R, 3, ISFJ). In addition, the majority emphasized cultivating the habit of engaging in physical activity, having perseverance when performing exercises, and reducing physical deterioration as their long-term aims: “The long-term goal is to exercise two to three days a week for one to one and a half hours each time” (F, R, 2, ESFJ); “My new goal is how to remain motivated to exercise” (F, R, 17, ISFJ); “I hope I can practice Tai Chi every day, I am satisfied if my body deterioration is slowed down” (F, T, 6, ISFJ). Moreover, one participant (F, T, 12, ESFJ) believed that achieving long-term health-related goals needed time for learning and building up: “It needs time to build up. I want to further learn Tai Chi stances, change the eating pattern and focus on improving healthy lifestyles”. 

Importantly, a female participant (F, R, 18, ESFJ) mentioned that health is not the same as longevity and that being able to live independently is critical: “The goal is not to live a long life, but to be able to walk even at an advanced age, without the need for assistance or difficulty in walking”.

Finally, a participant said that a further goal was not only to promote health for herself but also to encourage others: “The goal is now to encourage other people to do exercise together, not just myself” (F, R, 11, ESFJ).


**Goal preference for intervention/program: generalized vs. personalized aims**


Participants’ opinions were mixed, with 10 preferring generalized aims, 9 preferring personalized aims, and 1 saying both were important. 

Participants believe generalized aims make them more motivated, encourage them, make them happy, and make teaching easier: “It is motivating to work together as a team towards a goal” (M, R, 1, ESFJ); “Generalized aims are more encouraging” (F, T, 9, ESFJ); “We are happier to have generalized aims because they can last for a longer time” (F, R, 17, ISFJ); “The course was too short with only 4 months. It would be too difficult for the instructor to get to know everyone in a short period of time if there was another opportunity to set up personalized aims for each person in the future” (F, R, 7, ESTJ); “Prefer generalized aims, because the instructor doesn’t know everyone’s abilities and physique” (M, T, 4, ISFJ). One participant said that generalized goals can protect their self-esteem: "Personalized goals disguised as one-to-one can be inferior or embarrassing. It’s embarrassing to see that others can but you can’t, so it’s better to have the same goal but give modifications at the same time” (F, T, 14, ESFJ). 

On the other hand, one participant preferred personalized aims since they were more in line with individual needs: “There was a big difference in the physical ability of the group for this program, so it would have been boring if I had to do the movements of the less fit students” (F, R, 15, ESFJ).


**Goal preference for intervention/program: ultimate aim vs. progressive small aims**


There were 14 participants who supported progressive small goals, whereas only 5 preferred ultimate goals, and 1 person chose both.

Progressive smaller goals will give participants a sense of success and satisfaction: "Smaller goals are more likely to have a sense of success and will be more intentional, large goals take a long time to see results. I am worried that something else is going on in the meantime that will affect the results, therefore less motivation and commitment” (F, R, 10, ISFJ). Some participants believe that small aims can make it easier for them to control their progress and are more suitable for their age: “For example, if I can’t achieve it today, I will try the next day and it will be easy to know if I have done it” (F, R, 15, ESFJ); “I can know the progress in each class and if go in the right direction” (F, R, 18); “It is better to take a gradual approach as you are getting older” (F, T, 5, ESFJ). 

Other participants believe that ultimate aims are more flexible: “Because we may not master the movements she [the instructor] taught us in each class, I hope that we will be given a target in the end, because if we have a target in every class, we will be under pressure and we will be afraid of being compared to others. I would rather have a long-term goal so that we can go home and practice even if we don’t do well in that class, and a small goal like asking us to hand in our homework”(F, R, 19, ESFJ).


**Personality, goals, and healthy lifestyle behavior**


In addition to filling out personality questionnaires, participants were also asked to describe their personalities. Those categorized as ESFJ, who comprised the majority, described themselves as positive, extroverted, agreeable, open, and conscientious. They like taking the initiative and trying new things, are open-minded about life, and enjoy team activities.

“I am open-minded, and always find activities and sports to participate in” (F, T, 9, ESFJ). “I like to interact and share with others, so I want to find activities and meet other people” (F, T, 12, ESFJ). “I learn spontaneously, and if it is a sport I am interested in, I will participate in it” (F, T, 14, ESFJ). “In the sports program, I was responsible for creating a WhatsApp group to keep my group members in communication” (F, T, 16, ESFJ). 

Those categorized as ISFJ described themselves as introverted, curious, anxious, and impatient as well as open-minded to changing themselves to maintain a healthy lifestyle.

“I am a person who likes to get to the bottom of things. I want to find theories to prove, to accept different opinions and then synthesize” (M, R, 3, ISFJ). “After participating in the program, I made new friends, contact with people makes me happier, let the introverted self become extroverted” (M, T, 4, ISFJ). “Actively looking for activities/groups sports can make myself cheerful … avoid overthinking” (F, R, 11, ISFJ). “Originally, I am a person who is in a hurry, but by participating in Tai Chi, I can slow myself down” (F, T, 6, ISFJ).

Only two participants fell into the INFP and ESTJ categories. The participant assessed as INFP described herself as passive and overthinking, which hindered her from connecting with others. The participant assessed as ESTJ believed she was an active, positive, self-disciplined person, which has a positive effect on maintaining a healthy lifestyle. 

### 5.3. Motivations and Intentions 


**Theory-based program**


The participants reported that being given regular health information was helpful, including the introductory definitions of physical and mental health, how to exercise, how to prevent injuries and reduce the risk of falling, how to eat healthily, etc.: “Giving participants advice on what level of physical or mental health they have, then explaining the means to them, and increasing their awareness of their own health” (F, T, 12, ESFJ); “Distributing exercise practices in the group, such as the movements, let us review, otherwise we will forget” (F, R, 15, ESFJ); “Holding health seminars on how to prevent injuries rather than only sport, provide us with knowledge such as fall prevention, injury triggers, diet and so on” (F, R, 18, ESFJ).


**Exercise encouragement**


Participants reported that verbal encouragement, coupons, coach supervision, and social support inspire them to maintain exercise. “Verbal compliments with a few material rewards” (F, R, 11, ESFJ). “Having coaches who can correct movement mistakes and encourage them to keep doing it” (F, T, 14, ESFJ). “Group sports can encourage and help each other” (F, R, 18, ESFJ).


**Perceived constraints on exercise**


Participants report that due to the COVID-19 pandemic, they had to stop all exercise programs. When the interviewer asked whether they used Zoom as a platform to participate in online exercise programs, some of them said that they did not because of privacy, limited space at home, or family issues. They also said that social interactions in face-to-face programs cannot be replaced by Zoom. Moreover, a few participants said that they could not adhere to exercise without a regular exercise program due to family, social matters, lack of discipline, and inertia. 

“Because of the pandemic, my exercise classes were stopped, and I’m lazy so I don’t do any exercise anymore” (F, R, 2, ESFJ).

“Using Zoom is not good enough, because of the different housing environment” (M, T, 4, ISFJ). “Zoom class is actually fine, but it’s a little more scattered because at home you’re thinking about other things and it’s hard to concentrate” (F, R, 8, ISFJ). “Because Zoom classes can only be conducted at home, there are no sports facilities at my home, especially full body mirror, I cannot see whether my movements are correct, however in the face-to-face program, I can do exercises with familiar classmates, and we can rectify each other’s mistakes in performing exercises, which cannot replaceable by Zoom” (F, R, 20, INFP); she added, “I am afraid I am not able to use the recording function in Zoom”.

“I am not very disciplined. Without an exercise program my exercises become irregular, some time watching TV, some time talking on the phone, sometimes shopping for food and cooking” (F, R, 15, ESFJ).

Table 1 and Table 2 provide summaries of the retrieved literature and the categories enriched by the literature and interviews. 

## 6. Discussion

The current review summarizes the interaction between healthy lifestyle behaviors, goal setting, and personality, and the results are supported and extended by interviews with participants. The study results demonstrate that healthy lifestyle behaviors mainly consist of participating in physical activity, adopting healthy eating habits, and engaging in social activities. There is a tendency for people with different personality traits to have different attitudes toward engaging in HLB. In addition to progress in achieving goals, both the literature review and interviews provide evidence for the use of different types of goal setting plans, including personalized goals, short-term and long-term goals, generalized goals, and progressive goals. 

Based on the review of the literature and interviews with participants, conscientiousness and neuroticism are the two personality traits that showed a significant interaction with personality, goals, and HLBs. Among those studies that focused on it, conscientiousness was shown to have a significant effect on the intention to protect one’s health, as well as the self-reported practice of health-protecting activities (e.g., exercise), according to the theory of planned behavior [61]. Furthermore, a study that used electronic devices in a physical activity intervention revealed that people with a high level of conscientiousness showed greater improvement in their daily steps, while people with health neuroticism and who have high levels of both neuroticism and conscientiousness had much higher daily steps [30]. It can be presumed that people with both neuroticism and conscientiousness have a strong fear of their possible future self, thus forcing them to develop greater self-regulation and self-control in order to prevent negative health-related outcomes, and it makes them more eager to strive for HLBs. However, the level of conscientiousness was shown to be changeable [62,63]. Nonetheless, studies documented that one’s level of conscientiousness increases with age [64]; hence, we acknowledge that most of the literature reviewed in the current study and the interviews with older adults tended to focus on the characteristic of conscientiousness. Studies have further indicated that a high level of conscientiousness is associated with more beneficial health behaviors and health-related outcomes [65,66]. Conversely, it is important to note that harmful behaviors such as drug and alcohol abuse do not seem to improve with conscientiousness over time or with age [67,68]. Therefore, this provides further evidence that conscientious people tend to take preventative actions to achieve health protection outcomes. Aside from age, social environment, and personal experience, psychological interventions such as behavioral cognitive therapy, mindfulness, and mental contrasting were found to be effective in increasing people’s level of conscientiousness by enhancing their commitment to goals and improving effective goal selection and goal striving, thus cultivating behavioral changes [63]. Based on this, interventions that include both physical activity and psychological elements should be promoted to cultivate conscientiousness and goal setting not only in older adults specifically but also in people in general, thus resulting in better HLB engagement. 

## 7. Future Implications

It is shown that extroverts and conscientious people are more capable of achieving goals, adjusting their goals, and re-engaging with goals compared to people with neuroticism; however, the effect of personalities on the types of goals preferred by older adults has been scarcely discussed. While the reviewed literature pointed out that the use of an electronic device during exercise, such as a Fitbit, can cultivate personalized goal setting by using the device’s planning, regulation, and monitoring functions, the effect of personalized goals showed a greater effect among people who were conscientious and introverted [30,48]. This was also indicated in the participants’ interviews. The interviews indicated that introverts mostly avoid social gatherings or group activities because they would like the activity to be individualized. Attending group activities, including physical activity interventions, can cause worry and anxiety, such as worrying about not being as capable as others or not being able to meet the standard of the group. On the contrary, extroverts are more likely to attend group activities, because they like to share their experiences with others or within the group, thus they prefer non-personalized goals. In addition, no discernible personality differences were found between the preference for ultimate goals and progressive goals. 

Based on the interview content, it is suggested that for physical activity or other HLB interventions in the future, the personality should be considered as a covariant and be controlled, or its possible effects on intervention outcomes should be examined during data analyses. Interventions that highlight individual differences could be offered, for instance, by recruiting participants with a particular personality type to improve their goal-setting skills and HLBs. Furthermore, different types of goals should be included in HLB inventions to investigate their effects on different personality traits when cultivating HLB. These suggestions are not exclusive to older adults but are also applicable to different populations and age groups. In addition, with an understanding of the importance of goal setting for healthy lifestyle behaviors, coaches and other health professionals can be given practical suggestions to enhance older adults’ self-understanding and literacy regarding healthy lifestyle behaviors (i.e., help them better understand their own physical capability, provide health information, or encourage them to set goals for themselves) before participating in exercise classes or undergoing medical treatments.

## 8. Strengths and Limitations of the Study

In the current study, we reviewed and summarized the interaction between goal setting, personality, and HLBs in the literature, further supported by qualitative interviews. Based on this review, it was found that conscientiousness, neuroticism, and extraversion had a significant effect on health-related behaviors as well as actual healthy lifestyle behaviors. Although personality traits such as agreeableness and openness were less likely to be examined or showed no significant outcomes, from the qualitative interviews, we are able to further conclude that older adults with all types of personalities, except neuroticism, tended to take the initiative to try new things, participate in physical activity interventions, and learn new knowledge. They were more open-minded toward unknown life and health-related knowledge, which facilitated their development and maintenance of a healthy lifestyle. On the other hand, older adults with neuroticism tended to overthink the unknown in the form of, for example, fear over their possible self or anxiety about the outcomes of new physical activities, thus holding them off from engaging in healthy behaviors. 

Given that the literature review was not a systematic review and might have neglected the quality of retrieved studies, it also could not indicate and quantify the interaction effects between personalities, goal setting, and healthy lifestyle behaviors. Moreover, the interviewees were recruited by using convenience sampling from a previous physical activity intervention; hence, it could be assumed that they have already engaged in regular HLBs. This represents a limitation in that we were not able to distinguish potential differences between older adults who do not engage in regular HLBs and those who are not willing to engage in group-based physical activity interventions. Additionally, in the current study, we adopted the Myers–Briggs Type Indicator (MBTI) instead of the Big Five Inventory, because we considered the difficulty of distinguishing the participants’ personalities based on the latter. This indicates the interaction effect between two or more strong traits within an individual. The interaction effect between personality traits can make it difficult to identify the specific effect of personality traits on HLBs and goal setting. Therefore, we adopted the MBTI, with the understanding that the Big Five Inventory would be more suitable for quantitative analyses. Therefore, a meta-analysis of the interaction between personalities, goal setting, and physical activities or other specific HLBs is suggested. In addition, examining different types of HLB interventions that consider personality and goal setting strategies in future research is highly encouraged. 

## Figures and Tables

**Figure 1 geriatrics-07-00131-f001:**
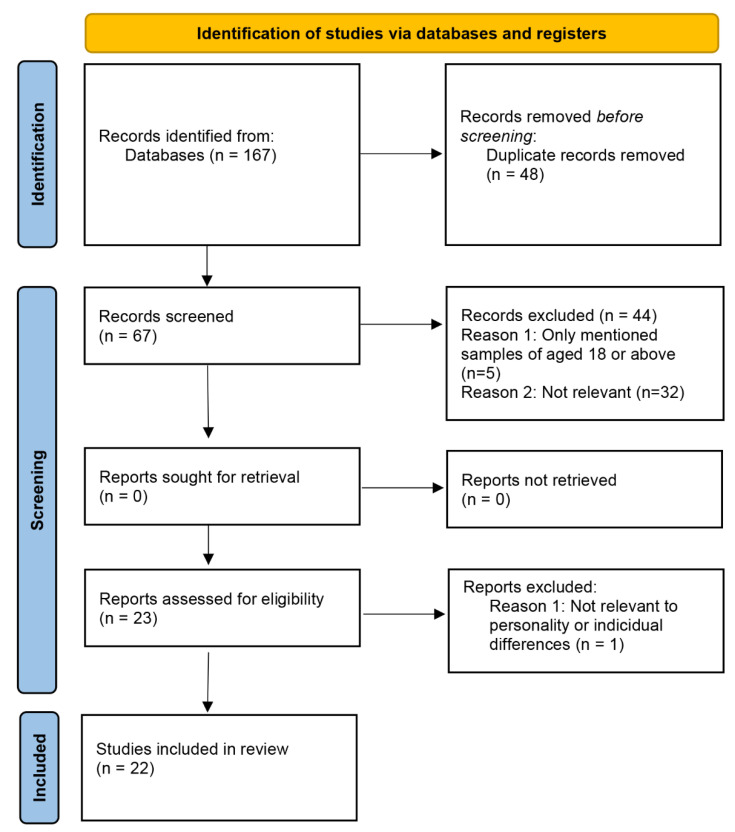
Flowchart of the literature search [18].

**Table 1 geriatrics-07-00131-t001:** Summary of the retrieved literature.

Authors	Year	Age	Population	F (%)	M (%)	Study Design	Purpose
Rapkin and Fischer	1992	mean = 73.3	n = 179	23.20%	76.80%	Cross-sectional study/scale	Examined how individual differences among elders influence goals in key life domains
Resnick, Orwig et al.	2005	mean = 80.9	n = 70	100%	N/A	Naturalistic or constructivist inquiry with single open-ended interview	Investigated how older women with hip fractures reacted to a motivational intervention designed to increase adherence to exercise
Churchill and Jessop	2010	mean = 33.05	n = 256	79.36%	20.64%	Prospective study	Investigated whether impulsivity moderated any effects of self-initiated implementation on avoiding snacking
Hankonen, Vollmann et al.	2010	50–65	n = 385	74%	26%	Longitudinal study	Studied whether modifiable and domain-specific social cognition, relatively stable and broad personality traits are the best predictors of waist circumference changes
Lethbridge, Watson et al.	2011	mean = 26.02 mean = 31.42	E: n = 238 C: n = 248	100%	N/A	Case–control study	Studied role of perfectionism, shape and weight overvaluation, dichotomous thinking, and conditional goals in psychopathology of eating disorders
Watson, Raykos et al.	2011	mean = 26.33	n = 201	100%	N/A	Cross-sectional study	Examined mediators of relationship between perfectionism and eating disorder psychopathology in a clinical sample
Palmer, Martin et al.	2014	S: mean = 50.1 H: mean = 50.7 (SD: 11.6)	Schizophrenia: n = 72 Healthy: n = 64	S:54.2% H:62.5%	S:45.8 H:37.5	Case–control study, interviews, and review of available medical records with SAPS/SANS/CES-D/BSI-A scale	Studied happiness in 72 outpatients with non-remitted chronic schizophrenia with a mean duration of 10 years
Smagula, Faulkner et al.	2016	mean = 81.4 (SD: 5.04)	n = 613	N/A	100%	Cross-sectional study	Evaluated relationships between specific personality factors and health
Kahlbaugh and Huffman	2017	65–97	n = 49	65.31%	34.69%	Cross-sectional study/scale	Examined personality traits of older adults and their emotional experiences associated with engaging in specific leisure activities
Smith, Williams et al.	2017	21–65	n = 94	68.08%	31.92%	Cross-sectional study	(i) Determined if Big Five personality traits are associated with physical activity over two weeks; (ii) assessed whether social-cognitive constructs predict physical activity; (iii) compared participants at various activity levels regarding differences in social-cognitive constructs and personality factors; (iv) examined the impact of personality and social cognitions on physical activity behavior
Briki	2018	18–65	n = 501	58.70%	41.30%	Cross-sectional study	Investigated relationships between personality, LTPA, and feelings
Jarrahi, Gafinowitz et al.	2018	NA	n = 29	65.52%	34.48%	Cross-sectional study	Explored how different types of pre-existing motivation shape people’s perception and adoption of the device
Javaras, Williams et al.	2019	N/A	N/A	N/A	N/A	Systematic review	Overview of existing cognitive behavioral, metacognitive, and cognitive remediation interventions to influence conscientiousness, providing several suggestions
Briki and Dagot	2020	mean = 33.63	n = 418	66.70%	33.30%	Pilot study: 10-item Flow Short Scale/ questionnaire	Presumed that conservatives are happier than liberals because they develop better mental adjustment especially under contextual threat
Davis, Murphy et al.	2020	45–75	n = 28	N/A	N/A	Pre-post pilot study	Evaluated pilot 12-week physical activity and diet program delivered by virtual assistant
Kappes and Thomsen	2020	Study 1: mean = 32.65 Study 2: mean = 25.9	Study 1: n = 67 Study 2: n = 60 * n-couples	N/A	N/A	Cross-sectional study	Analyzed whether viewing a partner’s regulatory technique would encourage imitation when faced with similar problems pursuing a goal and translated to overcoming challenges in a different setting; also evaluated impact of boundary conditions on imitation of goal regulatory processes
Stieger, Robinson et al.	2020	35–69	n = 52	71.20%	28.80%	Cross-sectional study	Examined whether personality traits predict the extent to which people increase their number of daily steps over 35 days
Bischoff, Baumann et al.	2021	mean = 47.7	n = 1008	59%	41%	Web-based survey	Identified requirements for a tailored app to reduce stress in a cohort of highly stressed families
Kim, Nho et al.	2021	40–60	E: n = 31 C: n = 32	100%	N/A	Randomized control group pretest–post-test	Examined lifestyle changes for low-income middle-aged women to actively manage risk factors
Sirois	2021	mean = 34.03	n = 191	67.50%	32.50%	Prospective cohort study	Investigated association between procrastination and self-regulatory processes as reflected in efficacy
Allen, Gmelin et al.	2022	mean = 84.3	n = 1670	N/A	100%	Cross-sectional study	Examined association between personality measures and perceived mental fatigability
Zimmermann and Chakravarti	2022	35–44	n = 281	44%	56%	Dilution effect studies	Investigated whether people who are physically active on a regular basis make different decisions than people who are not
		mean = 36	n = 289	52%	48%		
		mean = 38	n = 120	61%	39%		
		45–49	n = 268	49%	51%		
		mean = 34	n = 257	47%	53%		

**Table 2 geriatrics-07-00131-t002:** Summary of categories and coding for literature synthesis and interviews.

Category	Source	Subcategories	Detailed Information
Healthy lifestyle behaviors	Literature and interviews	Physical exercise (using electronic devices)	“Planned, structured, repeated and with maintenance” [21], with intention and beliefs
	Literature	Healthy eating habits	Weight control and nutrition management
		Engaging in social activities	/
Goal setting	Literature	Types of goals	Achievement, maintenance, disengagement/engagement, and compensation goals [47]; preventative and personalized goals for “hoped-for” possible self and “feared-for” possible self
	Interviews	Types of goals	Personalized, generalized, ultimate, and progressive goals
Personality	Literature	Big Five personality traits	Openness, conscientiousness, extraversion, agreeableness, neuroticism
	Interviews	Optimism	Likes to make friends, engage in social activities
		Introversion	Curious, anxious, and impatient but open-minded

## Data Availability

Not applicable.

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
