# Peer review of "Healthy Lifestyle Behavior, Goal Setting, and Personality among Older Adults: A Synthesis of Literature Reviews and Interviews"

_geriatrics, 2022, doi:10.3390/geriatrics7060131_

Round 1

Reviewer 1 Report

Dear author, 

Congratulations on the presentation of this article.

Secondly, I would like to suggest some recommendations to improve the interest in your article, downloads, quotes, greater internationalisation.

In the introduction, and conceptualising a health issue, I advise you to better conceptualise the variables that can influence and that we should not forget. That is why I suggest that in the introduction, in line 44 I suggest you expand the introduction, referring to variables that can influence the practice of physical activity, which are contextual, but which we cannot forget with recent quotes:

"To better understand the reasons for physical activity practice in older people, we must take into account socio-contextual variables that determine a greater or lesser practice such as gender, educational level, level of functional capacity and the practice of leisure activities, with women specifically and a low educational level being the elements in ageing that hinder a proactive attitude towards good capacities and maintaining a good level of leisure activities in older people" https://doi.org/10. 3390/bs12090331 "Likewise, socio-environmental factors related to a higher quality of life also influence a greater practice of physical activity in older people, such as income and education, with training in healthy lifestyle habits being one of the pillars for acquiring healthy lifestyle habits, such as practising greater physical activity in older people" https://doi.org/10.3390/ijerph182010815.  "In fact, older people value that among all the elements that make up their quality of life and benefit them, health care, relationships, functional autonomy and staying active, and therefore, physical activity is one of the most important elements in the life of the older person and in their quality of life" https://dialnet.unirioja.es/servlet/articulo?codigo=6477858.

At the end of your discussion I suggest: 

-Include: Practical implications of research on older people.

-Include: Theoretical implications for other researchers or academics.

-Include: Limitations.

-Include: strengths of your study compared to others.

-Include: Future lines of research.

Congratulations on the study and welcome these positive suggestions in order to improve your internationalisation, downloads and citations. 

Best regards and all the best.

Reviewer 2 Report

Dear authors,

Congratulation on your interesting article. In general, it is very interesting and quite trendy. 

Here are my suggestions:

Regarding the objectives, these are not clear. Although you have presented 2 main research questions, these can be answered in differnt forms. So it is necessary to determine specifically what is your research aim. I suggest writing the aim at the end of the introductions/background as usually is done in most articles.

According to the method, it seems you have used the PRISMA statement. However, you have not cited it. It is only referenced at the end of your work. 

With regard of the results, I suggest the use of tables to show the literature review resume.

Kind regards
